# *SpermQ*–A Simple Analysis Software to Comprehensively Study Flagellar Beating and Sperm Steering

**DOI:** 10.3390/cells8010010

**Published:** 2018-12-26

**Authors:** Jan N. Hansen, Sebastian Rassmann, Jan F. Jikeli, Dagmar Wachten

**Affiliations:** 1Institute of Innate Immunity, Biophysical Imaging, University Hospital Bonn, University of Bonn, 53127 Bonn, Germany; jan.hansen@uni-bonn.de (J.N.H.); sebastian.rassmann@gmx.de (S.R.); jf.jikeli@gmail.com (J.F.J.); 2Center of Advanced European Studies and Research (CAESAR), Molecular Physiology, 53175 Bonn, Germany

**Keywords:** cilia, flagella, beat pattern, motility, navigation, sperm

## Abstract

Motile cilia, also called flagella, are found across a broad range of species; some cilia propel prokaryotes and eukaryotic cells like sperm, while cilia on epithelial surfaces create complex fluid patterns e.g., in the brain or lung. For sperm, the picture has emerged that the flagellum is not only a motor but also a sensor that detects stimuli from the environment, computing the beat pattern according to the sensory input. Thereby, the flagellum navigates sperm through the complex environment in the female genital tract. However, we know very little about how environmental signals change the flagellar beat and, thereby, the swimming behavior of sperm. It has been proposed that distinct signaling domains in the flagellum control the flagellar beat. However, a detailed analysis has been mainly hampered by the fact that current comprehensive analysis approaches rely on complex microscopy and analysis systems. Thus, knowledge on sperm signaling regulating the flagellar beat is based on custom quantification approaches that are limited to only a few aspects of the beat pattern, do not resolve the kinetics of the entire flagellum, rely on manual, qualitative descriptions, and are only a little comparable among each other. Here, we present SpermQ, a ready-to-use and comprehensive analysis software to quantify sperm motility. SpermQ provides a detailed quantification of the flagellar beat based on common time-lapse images acquired by dark-field or epi-fluorescence microscopy, making SpermQ widely applicable. We envision SpermQ becoming a standard tool in flagellar and motile cilia research that allows to readily link studies on individual signaling components in sperm and distinct flagellar beat patterns.

## 1. Introduction

Reproduction starts with fertilization, the fusion of sperm and egg. In mammals, sperm need to pass a complex environment, the female genital tract, to reach the site of fertilization. On their way through the female genital tract, sperm cells are guided by different physical and chemical cues. Here, the flagellum functions as both a sensor and motor, allowing guided motility to the site of fertilization [1]. Several physical and chemical cues guide the sperm cell through the complex environment in the female genital tract. Chemotaxis, thermotaxis, haptotaxis, and rheotaxis have been described to control the sperm’s swimming path [1]. The flagellum transduces this sensory input from the environment into a distinct flagellar beat pattern [2]. A symmetrical flagellar beat results in a straight swimming path of the sperm cell, whereas asymmetries in the beat pattern lead to a curved or even helical swimming path [3,4,5,6,7]. In fact, to steer in gradients of sensory cues, sperm adjust the flagellar waveform. Thus, control of beat asymmetry in sperm is a key regulator of steering and navigation. In sea urchin sperm, the navigation principles are well established: In a 2D gradient, sperm swim on looping trajectories; in a 3D gradient, the swimming helix bends to align with the gradient while the symmetry of the flagellar beat pattern determines the swimming path of the sperm cell [2,5,8]. However, the navigation of mammalian sperm is less well understood. In bull sperm, an asymmetric flagellar beat pattern has been directly correlated to a curved or rotating swimming path [3,6,7]. In human sperm, analyzing the beat pattern in 3D revealed that during rheotaxis, sperm swimming on a curved trajectory display an asymmetric beat in the midpiece [9]. In addition, human sperm steer with second harmonics of the flagellar beat [10]. The beat pattern is characterized by two bending waves—one set by the fundamental frequency and the other by its second harmonic. The second harmonic breaks the symmetry of the waveform and contributes to steering [10]. Many studies unraveled only selected aspects of the flagellar beat, investigating only a small part of the flagellum [11,12], or analyzing the swimming path and swimming velocity only [13,14,15,16]. In addition, the current perspective on sperm signaling points towards a compartmentalization of signaling along the flagellum, e.g., in the case of cAMP signaling [17,18].

Thus, a detailed analysis of the flagellar beat is needed to understand sperm navigation. To this end, we have developed SpermQ, an analysis software that allows to comprehensively study the flagellar beat pattern and sperm steering using microscopy techniques that can be readily applied by almost any lab, i.e., dark-field and epifluorescence microscopy. SpermQ cannot only analyze the beat of tethered sperm, but also the beat of freely-swimming sperm. We envision SpermQ becoming a standard tool in sperm research.

## 2. Materials and Methods

### 2.1. Mouse Sperm

Wild-type mice (C57BL/6) were obtained from Janvier Labs (Le Genest-Saint-Isle, France). Animal care and experiments were in accordance with the relevant guidelines and regulations and approved by the local authorities (Landesamt für Natur, Umwelt und Verbraucherschutz, North Rhine-Westphalia, LANUV). Mice were killed by cervical dislocation after isoflurane (Curamed Pharma, Karlsruhe-Durlach, Germany) inhalation. Sperm were isolated by incision of the cauda epididymis followed by a swim-out in modified TYH medium (in mM: 135 NaCl, 4.8 KCl, 2 CaCl_2_, 1.2 KH_2_PO_4_, 1 MgSO_4_, 5.6 glucose, 0.5 sodium pyruvate, 10 lactic acid, 10 HEPES, pH 7.4 adjusted at 37 °C with NaOH). After 15–30 min swim-out at 37 °C, sperm were collected and counted.

### 2.2. Human Sperm

Human semen samples were donated by healthy adult males with their prior written consent and the approval of the ethic committee of the University of Bonn (042/17). The sperm cells were purified by a “swim up” procedure [19] using human tubular fluid (HTF) (in mM: 97.8 NaCl, 4.69 KCl, 0.2 MgSO_4_, 0.37 KH_2_PO_4_, 2.04 CaCl_2_, 0.33 Na-pyruvate, 21.4 lactic acid, 2.78 glucose, 21 HEPES, and 25 NaHCO_3_ adjusted to pH 7.3–7.4 with NaOH).

### 2.3. Imaging

Dark-field imaging was performed at an inverted microscope (IX71; Olympus, Hamburg, Germany) equipped with a dark-field condenser and a high-speed camera (PCO Dimax, Kehlheim, Germany). Image sequences of human sperm were recorded with 500 frames per second (fps) and using a 20× objective (NA 0.5, UPLFLN, Olympus, Hamburg, Germany); image sequences of mouse sperm were recorded with 200 fps using a 10× objective (NA 0.4, UPlanFL; Olympus, Hamburg, Germany) with an additional 1.6× magnifying lens (Olympus, Hamburg, Germany) that was inserted into the light path (final magnification: 16×). For imaging, the sperm solution prepared by the swim-up technology was inserted in a custom-made observation chamber with a depth of about 150 µm (Figure 1). The temperature of the microscope incubator (Life Imaging Services, Basel, Switzerland) was adjusted to 37 °C for imaging.

### 2.4. Image Analysis

All image processing and analysis was performed using ImageJ (v1.52i, National Institutes of Health, Bethesda, MN, USA). For images of mouse sperm, a minimum-intensity projection of each image sequence was generated and subtracted from the respective image sequence. Thereby, background signals were reduced in the image. In images of human sperm, the background was removed by the ImageJ (v1.52i, National Institutes of Health, Bethesda, MN, USA) function subtract background (radius: 10 px for a magnification of 20×). After background correction, image sequences of mouse and human sperm were subjected to SpermQ analysis (settings: Table 1).

### 2.5. Analysis Workflow Underpinning SpermQ

SpermQ is a fully automated analysis software to quantify sperm navigation and the flagellar beat of sperm. SpermQ was developed as an ImageJ plugin (v1.52i, National Institutes of Health, Bethesda, MN, USA) in Java. We will present the workflow performed by SpermQ in the following. The parameters applied in the SpermQ workflow are listed in Table 1 and will be referenced at the different steps. For each image sequence, each time frame is processed separately. To reconstruct the flagellum, the frame image is filtered by a Gaussian kernel with a σ defined by the user (Table 1B), binarized using a thresholding algorithm selected by the user (Table 1A), again Gaussian blur filtered with the same σ (Table 1B–C), and processed by the ImageJ (v1.52i, Bethesda, MN, USA) plugin Skeletonize3D [20]. The Gaussian blur can be restricted to the ROI set by the user, e.g., to blur only the sperm head while preserving the fine structure of the flagellum (Table 1D). The generated skeleton image is analyzed (AnalyzeSkeleton) [20] to retrieve the pixels of the skeleton.

For each pixel, a point $\vec{p_{i}}$ is generated using the coordinates of the pixel and all points are collected in a point list P (the index i indicates the position of the point in P; 0<i≤|P|, while |P| defines the number of points in the list). Next, P is sorted so that $\sum_{i=2}^{\left|P\right|}\bar{p_{i-1}p_{i}}$ is minimal. To ensure that $\vec{p_{1}}$ corresponds to the sperm head and not to the flagellar tip, we take advantage of the fact that dark-field images are brighter at the head, due to a stronger light scattering than at the tip of the flagellum. The average image intensities in circular ROIs (radius: 8 pixels) around $\vec{p_{1}}$ and around $\vec{p_{\left|P\right|}}$ are compared and the order of P is inverted if the average intensity around $\vec{p_{1}}$ is smaller than around $\vec{p_{\left|P\right|}}$. SpermQ can be set to add the center of mass of the circular ROI as the first point of P (Table 1F) or to set the first point to the same position in all frames (Table 1G). For the latter, $\vec{p_{1}}$ is set in each frame to the average position of all $\vec{p_{1}}$ in all frames across the time series. For a further description of the workflow, note, that the first points ($\vec{p_{1}}$, $\vec{p_{2}}$, $\vec{p_{3}}$) of P are localized in the sperm head as $\vec{p_{1}}$ corresponds to the center of the head.

Next, to obtain a fine flagellar curve, |P| is increased by linear interpolation between its elements: Between two consecutive points ($\vec{p_{i}}$ and $\vec{p_{i+1}}$), a distinct number of equidistant points is inserted. The number of points is set by the user (Table 1E); by default, two points are inserted between each pair of $\vec{p_{i}}$ and $\vec{p_{i+1}}$. Next, for each point $\vec{p_{i}}$, the tangential vector $\vec{t_{i}}$ pointing towards $\vec{p_{i+1}}$ is determined. The tangential vector $\vec{t_{i}}$ is determined by the vector between the points located a given number of points (Table 1I, divided by two) before and after $\vec{p_{i}}$. At the start or the end of P, closer points are selected to form $\vec{t_{i}}$. The normal vector $\vec{n_{i}}$ is determined as $\vec{n_{i}}=\left(-t_{i,y},{\;t}_{i,x}\right)$ for $\vec{t_{i}}=\left({\;t}_{i,x},{\;t}_{i,y}\right)$. If the cross-product $\vec{t_{i}}\times \vec{n_{i}}$ was negative, $\vec{n_{i}}$ was multiplied by −1. At each point of the flagellum, a line normal to the tangential direction and of a defined length was generated (Table 1J). Next, the intensity profile along the normal line was determined in steps of a pixel length (for a 20× magnification and an 11 µm camera pixel size: 0.55 µm/pixel). Each intensity value $i_{n}$ along the normal line is linearly interpolated from the four surrounding pixels weighted by their distance to the point on the normal line. The intensity profile is smoothed, if set so (Table 1L), and then fitted to a Gaussian curve using the CurveFitter implemented in ImageJ (v1.52i, National Institutes of Health, Bethesda, MN, USA) (maximum number of iterations: 1000). If the determined parameters of the Gaussian curve fulfilled all following four criteria, the point $\vec{p_{i}}$ was shifted by $c\cdot \vec{n_{i}}$ (c = center of the fitted Gaussian curve) to center $\vec{p_{i}}$ on the flagellum:
$r^{2}>0.8$ ($r^{2}$ = quality of the fit)a>0 (a = height of the Gaussian curve)d<2× normal radius for gauss fit (Table 1J) (d = width of the Gaussian curve)|c| < normal radius for gauss fit (Table 1J).

The points at the sperm head can be excluded from correction, if set by the user (Table 1K), which can improve the tracking of sperm with a head shape that does not fit well to a Gaussian curve.

Next, an arc length position $l_{p_{i}}$ for each flagellar point $\vec{p_{i}}$ as $l_{p_{i}}=\sum_{j=1}^{i}\left|\bar{\vec{p_{j-1}}\;\vec{p_{j}}}\right|$ is determined. However, as even small scattering of the points around the centerline can accumulate in erroneous $l_{p_{i}}$ results, especially at the flagellar tip, SpermQ removes outliers of the flagellar track and smoothens the track before determining $l_{p_{i}}.$ This procedure improves $l_{p_{i}}$ precision. Outliers from the reconstructed flagellar track are removed as follows: For each triplet of points ($\vec{p_{i-1}}$, $\vec{p_{i}}$, $\vec{p_{i+1}}$), $\vec{p_{i}}$ is removed from P if $\bar{\vec{p_{i-1}}\;\vec{p_{i}}}>\bar{\vec{p_{i}}\;\vec{p_{i+1}}}$. For smoothing, a custom-written algorithm is applied, which takes advantage of the fact that at a low distance range of a few micrometers, the flagellar track can be approximated by a line. In brief, lines are generated based on neighbored points of $\vec{p_{i}}\;$, and $\vec{p_{i}}$ is projected to the line that allows a median shift of $\vec{p_{i}}$ compared to shifting it to all other lines. The median and not the average was selected as this is less sensitive to individual off-range values. In detail, for a given point $\vec{p_{i}}$, a distinct number (Table 1O) of upstream and downstream points from $\vec{p_{i}}$ are included. At the start and the end of the point list, fewer points are included. Pairs of the included points are generated in all possible combinations. For each pair, a line between the paired points is constructed, and point $\vec{p_{i}}$ is projected to that line, resulting in the projected point $\vec{p_{i}}’$. The median distance $\vec{p_{i}}-\vec{p_{i}}’$ of all pairs is determined and the flagellar track point $\vec{p_{i}}$ is replaced with the projected point $\vec{p_{i}}’$ belonging to the pair whose projection distance is nearest to the median distance. Next, arc length positions are determined as described above. After smoothing, outliers, if still existing, are removed again and normal and tangential vectors are calculated as described. Again, normal lines are constructed and the Gaussian-curve fits on the normal line are recalculated. The points with fits that do not match the above described criteria can be removed, if set (Table 1H). For each point $\vec{p_{i}}$, the width parameter of the fitted Gaussian curve serves as a relative z position of the flagellum (see also results text). The relative z position values are smoothed along the flagellum analogously to the described method for the smoothing in xy, except that upstream and downstream points are only considered for smoothing if they are below a defined arc length distance to the smoothed point $\vec{p_{i}}$ (Table 1N). Additionally, the Gaussian curve height is retrieved and saved as an intensity value for each flagellar point (see results).

In the next step, all flagellar positions are translated into a new coordinate system whose x axis is defined by the vector $\vec{v}$ from the first point to the point at a set arc length position (Table 1P). All x and y positions in the new coordinate system are plotted as a function of the arc length in kymographs (see results). Additionally, the angle between the x axis of the image and $\vec{v}$ (angle θ) is generated. Further parameters are determined as described in the results part, e.g., the curvature angle is determined by the angle between the tangential vector at the given point and the tangential vector of the point in a defined arc length distance (Table 1Q) before the given point. While the sperm cell is rolling around its longitudinal axis, the intensity of the light that is scattered from the head to the direction of the objective during dark-field imaging oscillates and can be used to quantify the rolling of the sperm cell. To this end, SpermQ places a cross-sectioning line through the sperm head. The center of the line is defined by $\vec{p_{1}}$, the thickness of the line is fixed to 9 pixels and the length of the line is defined by the user (the length corresponds to twice the parameter called “head rotation matrix radius”, Table 1T). Intensity values on the line are sampled in steps of one pixel length. Intensity values at inter-pixel positions are linearly interpolated from the four surrounding pixels. For each step on the line, the intensity value is determined as the average of all intensities within the line-thickness range. The maximum intensity value on the line is generated and all maximum intensity values over time are subjected to FFT; in the resulting frequency spectrum, the primary frequency peak serves as the rolling frequency and the secondary frequency peak refers to the main beat frequency. Additionally, all other parameters are subjected to frequency analysis. Generally, the FFT algorithm in SpermQ can not only be applied to the entire time sequence of a parameter but also to defined time steps of the image sequence; when the FFT parameter, described in Table 1R, is lower than the number of frames in the analyzed time sequence, a window of the defined number of frames is slid along the time sequence and all FFT results are determined separately for each window. Optionally, the sperm head can be excluded from the frequency analysis of flagellar parameters. For example, when the sperm head is so tightly tethered that it is not moving at all (e.g., in special cases of tethered sperm) the frequency parameters cannot provide any relevant results at the sperm head. In these cases, the user can decide to not perform a frequency analysis of the points corresponding to $l_{p_{i}}$ below or equal to a set value (Table 1S), thereby, speeding up analysis. Note, that P starts in the center of the head, so that the first points of P are localized in the sperm head.

### 2.6. Software

Image processing and analysis were performed in ImageJ (v1.52i, National Institutes of Health, Bethesda, MN, USA). Plots and Figures were generated using GraphPad Prism (GraphPad Software, Inc., v6.07, La Jolla, CA, USA) and Adobe Illustrator CS5 (Adobe Systems, Inc., v15.0.0, San Jose, CA, USA). SpermQ (v0.1.7) and SpermQ Evaluator (v1.0.2) were developed and compiled using Eclipse Mars.2 (Release 4.5.2, IDE for Java Developers, Eclipse Foundation, Inc., Ottawa, Ontario, Canada).

### 2.7. Software Availability

SpermQ will be made accessible upon request. SpermQ Evaluator is publicly available [21].

## 3. Results

### 3.1. Analysis and Workflow Performed by the Software SpermQ

SpermQ is an automated analysis software that is used to comprehensively study the flagellar beat of sperm in time-lapse image sequences generated by dark-field microscopy. Initially, SpermQ determines the flagellar trace in every frame of the analyzed image sequence. First, the raw image (Figure 2A) is gauss-filtered, segmented and skeletonized to obtain a rough trace of the flagellum (Figure 2B). Next, along the detected flagellar trace, normal lines are constructed (Figure 2C). The image intensities along each normal line are fitted to a Gaussian curve. The detected flagellar trace is adjusted to the center of the Gaussian curve, as this defines the precise location of the flagellum in the image (Figure 2D). After reconstructing the flagellum, SpermQ determines head parameters (Figure 3A–C) and flagellar parameters (Figure 3D–I).

The head parameters describe the location and orientation of the sperm cell in its environment and allow for studying sperm navigation. Head parameters are determined using the initial points of the reconstructed flagellar trace. SpermQ defines the head of the sperm cell, representing the first point of the detected flagellar trace, as its position in space (Figure 3A). This allows to track the swimming path of the sperm cell over time and thus, to determine whether the sperm cell swims straight or on a curved path. The head angle in space (Θ, Figure 3B) reveals the orientation of the sperm cell. Θ is defined as the angle between the head-midpiece-vector and the x-axis of the image. The oscillation of Θ determines the main beat frequency. The maximum intensity in the head serves as a parameter to determine rolling of the sperm cell around its longitudinal axis (Figure 3C). Due to the elliptical and slightly asymmetrical geometry of the sperm head, the intensity of the light that is scattered from the head into the direction of the objective during dark-field imaging oscillates while the sperm cell is rolling, i.e., rotating around its longitudinal axis. Determining the head parameters over time allows to reveal the trajectory of the swimming path, the curvature of the trajectory, the rolling of the sperm cell around its own axis, and the main beat frequency of the sperm.

The flagellar parameters link the flagellar beat pattern to the swimming path and describe the location, intensity and local curvature of individual points on the flagellum. A position on the flagellum is defined by its arc length, which is the distance of the given point to the head on the stretched flagellum. In each frame, all points along the flagellum are translated into a new coordinate system, whose x-axis is defined as the vector through the head and the midpiece of the sperm (Figure 3D). Calculating the range of the x- and y-coordinates of the flagellum relative to the head over time reveal the flagellar envelope, which refers to the beat amplitude. Additionally, the symmetry of the flagellar envelope compared to the head axis is important as this controls the swimming path of the sperm cell. Furthermore, tangential angles are determined at each point on the flagellum (Figure 3E). To determine the location and propagation of waves along the flagellum, two different parameters were included, which describe the local curvature of the flagellum independently of the head axis: the curvature (Figure 3F) and the curvature angle (Figure 3G). The curvature at a given arc length is determined based on the equation for the geometric curvature and using tangential vectors at defined distances (e.g., 5 µm) before and after the given arc length. The curvature angle at a given arc length is defined as the angle between the tangential vector at and slightly before (e.g., 10 µm) the given arc length. To consider that the flagellar beat also shows a component that is vertical to the focal plane of the microscope, we implemented a method that qualitatively describes the position of the flagellum along the optical axis of the microscope, representing the z-axis. The focal position o along the optical axis in the specimen that is recorded by the microscopy set-up is, according to the lens equation, defined as o = (f · i)/(i−f), where i is the position of the conjugate image generated by a lens with a focal length f. In a microscopy system, f is mainly represented by the objective, determining the focal position in the specimen. When an object is located at the focal position, it appears sharp in the image that is recorded by the microscopy set-up. The more distant the object is to the focal position along the optical axis, the less sharp and the wider it appears in the recorded image. This approach has been proposed already in the 20th century to qualitatively describe the position of the flagellum on the optical axis, representing the z-position [7]. Recently, this method was applied to describe the relative z-position of the flagellum at a given point by the width of the intensity peak on the normal line of the given flagellar point [9]. Similarly, we included an algorithm into SpermQ that fits the intensity profile of every normal line along the flagellum to a Gaussian curve, whose width parameter represents the z-position of the object (Figure 3H). The relative z-position, determined by SpermQ is not calibrated and does not reveal whether the object is located above or below the focal plane and, thus, the respective results have to be interpreted carefully. A way to solve this, however, has been published elsewhere [9], but has not been implemented in SpermQ.

To analyze fluorescently labelled sperm, SpermQ determines the fluorescence intensity at every arc length and in every frame as the maximum of the Gaussian curve fitted to the normal intensity profile (Figure 3I). This allows localizing fluorescent molecules in the flagellum and determining the fluorescence intensity along the flagellum. Of note, also the z position of the flagellum influences intensity measurements. However, this influence can be neglected when applying bright fluorescent labels or when filtering out intensity oscillations in the range of the flagellar beat frequency. For every flagellar parameter, SpermQ generates a kymograph which displays the parameter results intensity-coded as a function of arc length and time, depicting the whole kinetics of an individual parameter at every flagellar position.

The flagellar beat of the sperm cell is highly periodic and thus, can be considered as an oscillation. To determine the beat frequency of sperm, Fast Fourier Transformation (FFT) of the oscillating parameter and analysis of the resulting frequency spectrum is commonly used to determine the beat frequency and other beat characteristics, such as the phase of the oscillation or additional frequencies on the flagellum that control the swimming path [10]. To implement this method into SpermQ, we included a FFT algorithm (derived from the package edu.emory.mathcs.jtransforms.fft by Piotr Wendykier, Emory University). The FFT algorithm is used to translate the time course of each parameter, and for the flagellar parameters also at each flagellar position, into a frequency spectrum. For each frequency spectrum, SpermQ determines the positions of the highest (primary frequency) and the second highest (secondary frequency) peak in the spectrum. In addition, SpermQ determines the center-of-mass (COM) of the frequency spectrum. The COM describes the distribution and amplitude of all frequency peaks in the frequency spectrum. As the COM depends on the characteristics of the whole frequency spectrum, the COM serves as a quick and simple measure to indicate differences in the frequency spectra of multiple cells (or over time). Large differences in the COM highlight that the user should analyze the respective frequency spectra in more detail.

In summary, SpermQ determines the location and orientation of the cell in space, the precise beat pattern along the flagellum, the intensity levels on the flagellum, and the frequency characteristics of all of these parameters. An overview of all parameters is presented in Table 2.

### 3.2. SpermQ Analysis of the Beat of a Head-Tethered Mouse Sperm

To demonstrate the power of SpermQ, we analyzed dark-field microscopy images of a head-tethered mouse sperm. The orientation of the sperm cell to the head midpiece axis (Figure 4A,B) demonstrates that the flagellar envelope is symmetric. Analysis of Θ reveals that the cell beats regularly with a beating frequency of 11 Hz (Figure 4C) without rotating around the tethered point (the overall Θ level is stable over time). The homogeneity of the beat pattern is confirmed by the curvature angle kymograph (Figure 4D). Based on the kymograph, SpermQ detects a primary frequency peak of 11 Hz along the entire flagellum and a secondary frequency peak of 23 Hz close to the flagellar tip (Figure 4E, bottom panel). This shows that SpermQ is able to resolve the flagellar beat pattern spatially along the flagellum. The same flagellar beat frequencies are apparent in the kymograph that depicts the relative z position of the flagellum (Figure 4F). Of note, this kymograph displays multiple maxima/minima for one oscillation at the tip of the flagellum (Figure 4F). This is due to the lack of direction information compared to the focal plane in the parameter relative z position. When the flagellum crosses the focal plane during its beat cycle, a single beat will result in a double frequency peak. As a result, in the frequency analysis, the double frequency of the primary beat frequency is detected and adds on the power of the second harmonic frequency in the spectrum, resulting in the false detection of the second harmonic frequency as the primary frequency peak (Figure 4G). This example highlights that the relative z position needs to be carefully interpreted. However, this shortcoming can be avoided by setting the focal plane of the microscope in the cover glass adjacent to the tethered cell that is recorded.

### 3.3. SpermQ Analysis of the Beat of a Freely-Swimming Human Sperm

SpermQ is also able to analyze the beat and swimming path of freely-swimming sperm, e.g., in dark-field microscopy images of human sperm (Figure 5). Head-tethered sperm are not able to roll around their longitudinal axis, and thus, their flagellar beat is mainly parallel to the focal plane. In contrast, freely-swimming sperm roll around their longitudinal axis while beating in all different directions. The 2D projection of one beat cycle of freely-swimming human sperm displays an asymmetric shape of the beat (Figure 5A), even when aligning the sperm cell to the head midpiece axis in 2D (Figure 5B). Here, the rolling frequency is 9 Hz and the Θ results show that two periodic events determine the flagellar beat frequency measured in 2D (Figure 5B): a primary frequency of 9 Hz (which corresponds to the rolling frequency) and a secondary frequency of 27 Hz (= three times the primary frequency). This indicates that while the sperm is rolling once around its longitudinal axis, the flagellum performs three beat cycles. By tracking the head position, SpermQ creates trajectories of the swimming path (Figure 5D) and determines a precise kymograph of the curvature angle along a freely-swimming sperm (Figure 5E). Taken together, SpermQ precisely quantifies the beat of freely-swimming sperm.

### 3.4. SpermQ Is a Readily-Applicable Software

We developed SpermQ with the goal to create a software that is easy to handle, fast, allows batch processing, and requires as little user interaction as possible. SpermQ is a completely automated software. Before applying the software, it is recommended to split the image stack to be analyzed into single-sperm image stacks to exclude other particles (e.g., dirt particles, other sperm) from the analyzed image. This also increases the analysis speed by reducing the data size of the processed image. Each of the single-sperm image stacks needs to be cropped to the minimum required size but leaving a little bit of extra space (few microns) around the flagellum. The extra space is required for Gaussian curve fits along the flagellum (Figure 2C). Before loading these stacks into SpermQ, the background in the image has to be reduced. To this end, the “Subtract Background” method in ImageJ (v1.52i, National Institutes of Health, Bethesda, MN, USA) can be applied, which, when a low radius is selected in the “Subtract Background” settings, also equalizes intensity differences in the sperm cell, thereby improving flagellar reconstruction by SpermQ.

After preprocessing, the data can be readily applied to SpermQ. When launching SpermQ, a settings dialogue opens to set default settings for distinct types of data sets (e.g., depending on the sperm species or the magnification used). To load data into SpermQ, a dialogue containing an analysis list opens, to which the user can add single-sperm image stacks from the hard disk. After listing all sperm image stacks, the software can be started. In the first step, time projections of each selected image stack are presented, for which the user is asked to set a ROI. This ROI should contain an image region where the sperm cell is always present in the entire sequence. For tethered sperm, a circle around the head is sufficient. For freely-swimming sperm, a ROI around the sperm’s swimming path is set. Setting this ROI reduces the regions where SpermQ searches for a flagellum and, thereby, increases the analysis speed and precision of the software, as e.g., remaining dirt particles in the image will be neglected. After setting the ROIs, SpermQ analyzes all image stacks without further user interaction. The results are automatically saved into a newly created analysis folder, separately for each analyzed sperm.

To visualize the results, we developed an additional open-source java application, the SpermQ Evaluator [21]. In SpermQ Evaluator, data sets can be loaded and ordered by the user, and overview tables for the loaded data sets are created. In addition, SpermQ Evaluator saves an overview PDF for each analyzed data set, comprising a collection of plots of the most important parameters (see Appendix A).

## 4. Discussion

How mammalian sperm control their flagellar beat pattern and, thereby, their swimming path, has not been fully resolved. Studies that investigated the flagellar beat already started 50 years ago [22,23,24,25] and the molecular mechanisms underlying axonemal beating have been analyzed over the last years [26,27,28,29,30,31,32]. Some studies already quantified the flagellar beat in great detail [33,34,35]. However, for many labs, a detailed analysis of the flagellar beat pattern has not been possible due to the lack of suitable software and hardware solutions that are widely applicable. A plethora of custom-made imaging and image-analysis approaches have been used that were limited in the number of parameters or the fraction of the flagellum that could be analyzed, or were restricted to sperm velocity parameters only (e.g., computer-assisted sperm analysis (CASA)) [36,37,38,39]. Up to now, more than 12 different CASA systems are on the market. Most of them have established a centroid for each sperm cell and analyze the cell motility based on centroid trajectory [36]. These systems were optimized according to the needs of clinicians, who require a robust, simple system to analyze percentages of motile sperm (based on velocity parameters like VCL, VAP, and/or LIN) and sperm morphology. However, an analysis tool that allows an in-depth characterization of all cilia or flagella beat parameters was missing.

With SpermQ, we developed a software that analyzes the flagellar beat in detail while requiring only simple imaging approaches (dark-field or epifluorescence microscopy). SpermQ is not only suited to analyze images from dark-field microcopy, but, in principal, also DIC and phase contrast images. However, to subject these images to the analysis pipeline, they need to be more extensively pre-processed and the fitting regime needs to be changed. One point that has to be taken into account when recording images is that the Nyquist–Shannon sampling theorem has to be satisfied to accurately determine the flagellar beat frequency. This will be fulfilled when imaging at 500 fps (for human sperm) and can even be achieved by cameras that are not high-end.

SpermQ analyzes the flagellar beat in depth, unravelling different head and flagellar parameters, including a FFT-based frequency analysis of all parameters. In combination with the evaluation tool (SpermQ Evaluator), comprehensive results can be rapidly and clearly displayed. Furthermore, SpermQ is automatized, allowing to analyze large datasets and excluding the bias of user-dependent custom analysis approaches. By providing default settings, SpermQ is readily applicable by any user. The program can be used not only to track tethered sperm but also freely-swimming sperm. Thereby, the software is particularly well suited for labs that do not have the expertise and technical requirements to perform this complex analysis yet.

In particular, the software will allow to study the molecular mechanisms underlying sperm navigation. Four different mechanisms have been identified that guide sperm to the egg: chemotaxis, haptotaxis, thermotaxis, and rheotaxis [1]. Using genetically-modified model organisms and novel pharmacological tools, we are only beginning to understand the signaling pathways underlying these different sperm navigation mechanisms. We envision that SpermQ will readily improve the analysis, making it easy to perform by any user and, thereby, more comparable.

## Figures and Tables

**Figure 1 cells-08-00010-f001:**
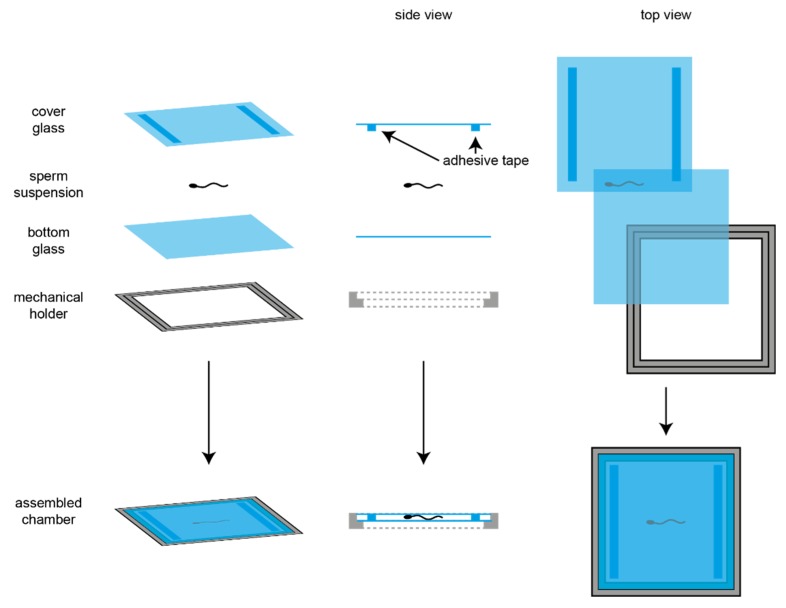
Custom-made observation chamber. The observation chamber was produced by a custom-made mechanical holder (metal), in which a glass cover slip was placed. Next, a drop of sperm in suspension was placed in the center of the cover slip. Then, another cover slip was placed on top, to which two piles of each three layers of adhesive tape (Tesafilm^®^, Tesa, Norderstedt, Germany) were placed, so that a chamber depth of 150 µm was generated (depth was measured).

**Figure 2 cells-08-00010-f002:**
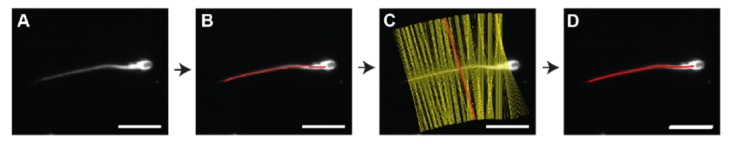
Reconstruction of the flagellum. (**A**) Each frame image is segmented and skeletonized to obtain (**B**) a rough track (red) of the flagellum; (**C**) along the flagellum, tangential vectors are determined to generate normal lines (yellow: normal lines; red: exemplary normal line). The intensity profile along each normal line is fitted to a Gaussian curve to obtain the center of the profile. The track points are then corrected by shifting them to the center, (**D**) resulting in a precise flagellar track (the precision of the track after correction is below the image resolution and thus, cannot be correctly displayed in the image). Bar: 20 µm.

**Figure 3 cells-08-00010-f003:**
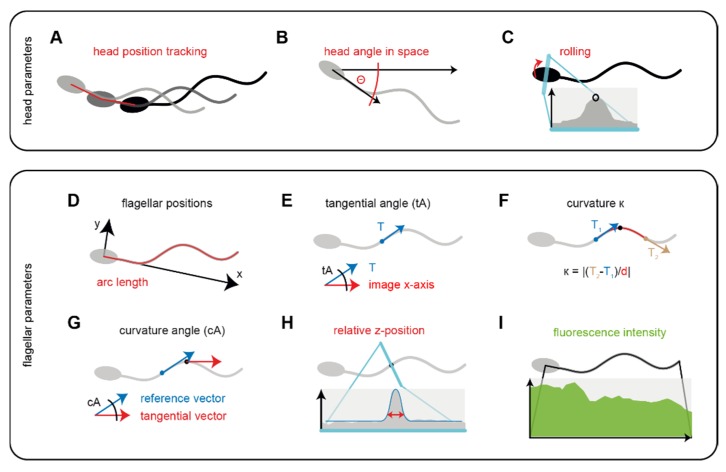
SpermQ output parameters. SpermQ output parameters characterize the orientation of the head in space (**A**–**C**) or each individual position on the flagellum (**D**–**I**). SpermQ outputs the head position in space over time (**A**) the angle Θ (**B**), which is the angle of the head midpiece axis to the x-axis. The maximum intensity (black circle) in a cross-sectioning line through the head center describes sperm rolling around its longitudinal axis (**C**). To determine the x- and y-positions of the flagellar points relative to the head, all flagellar points are translated into a coordinate system originating from the center of the head (**D**); the initial part (Table 1P) of the flagellum defines the x-vector of the coordinate system. Tangential angles are determined at each position as the angle between the tangential vector at the position and the x-axis of the image (**E**); the curvature (**F**) of a given point on the flagellum is determined as the geometric curvature and by using the tangential vectors (T_1_ and T_2_) at a defined distance d; d can be set by the user (see Table 1Q) and, here, was set to 10 µm (taking the points 5 µm up- and downstream into account for calculating the curvature). For arc lengths close to the head or the flagellar tip, a smaller distance is chosen. The curvature angle (cA) at a given point is calculated by the angle between the tangential vector at the given point and the tangential vector of the point at a defined distance (Table 1Q) before the given point (**G**); as a relative parameter for the flagellar z-position, SpermQ outputs the width of the Gaussian curve fitted to the intensity profile of the normal vectors at every point of the flagellum (**H**); furthermore, the maximum intensity of every flagellar point is measured to generate an intensity profile along the flagellum (**I**).

**Figure 4 cells-08-00010-f004:**
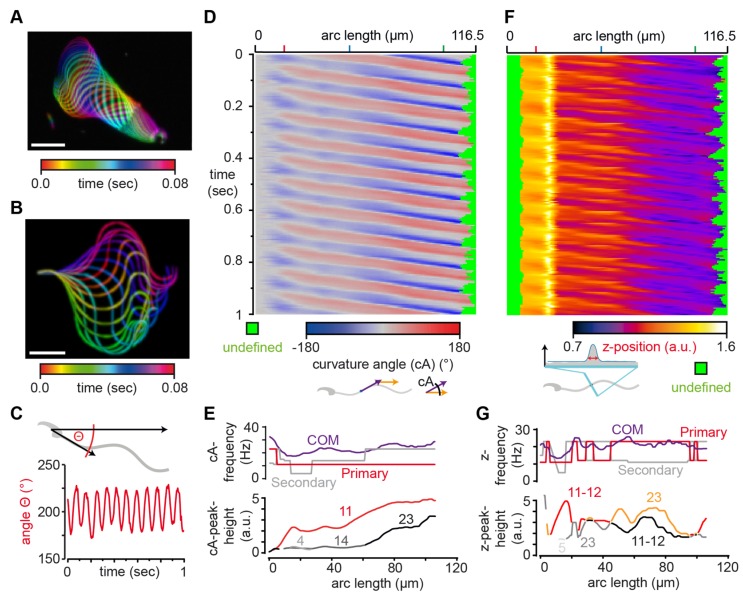
SpermQ reveals the beat pattern of a head-tethered mouse sperm. SpermQ analysis of a wild-type mouse sperm tethered to a glass surface at the head. Projection of the raw image (**A**) and the trace after orientation to the head midpiece axis (**B**), each representing one beat cycle; (**C**) SpermQ analysis reveals that the head orientation angle Θ stably oscillates with a primary frequency of 11 Hz around an angle of 200° in space (center of mass (COM) of frequency spectrum: 19.9 Hz; (**D**) kymographs of the determined curvature angle (cA) or (**F**) the z-position relative to the plane; (**E**) SpermQ frequency analysis of the cA or (**G**) of the z-position along the flagellum. Amplitudes for selected peaks in the frequency spectrum (frequency values indicated in Hz) are plotted in the bottom graph. Scale bars: 20 µm.

**Figure 5 cells-08-00010-f005:**
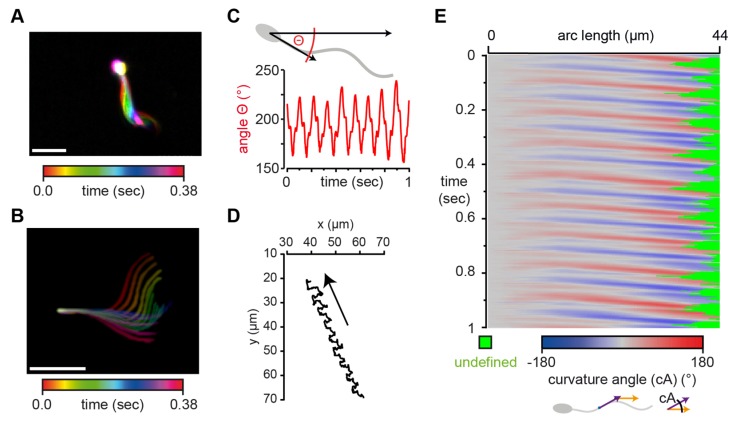
SpermQ reveals the beat pattern of a freely-swimming human sperm. SpermQ analysis of a human sperm, freely-swimming in a shallow observation chamber (depth 300 µm); (**A**) projection of the raw image and (**B**) the trace after orientation to the head midpiece axis, each representing one beat cycle; (**C**) SpermQ-analysis reveals that the head orientation angle Θ stably oscillates with the primary frequency of 9 Hz (rolling frequency of the sperm) and the secondary frequency of 27 Hz (beat frequency) (center of mass (COM) of frequency spectrum: 21.2 Hz); (**D**) trajectory of the sperm’s head position; (**E**) Kymograph of the determined curvature angle (cA). Scale bars: 20 µm.

**Table 1 cells-08-00010-t001:** SpermQ settings.

	Parameter	Mouse Sperm	Human Sperm
A	Thresholding Method	Li	Triangle
B	Gauss Sigma	3.0	2.0
C	Repeat gauss fit after binarization	false	true
D	Blur only inside ROI selection	true	false
E	Upscaling of point list (fold)	3	3
F	Add head center-of-mass as first point	false	false
G	Unify start points	false	false
H	Filter points by gauss fits	false	false
I	Maximum vector length (points)	14	20
J	Normal radius for gauss fit (µm)—defines the distance of each flagellar point to the start and end of the corresponding normal line = half-length of each normal line	5.0	6.0
K	Exclude head from correction or deletion	true	false
L	Smooth normal for XY gauss fit	true	true
M	Save Roi-sets or vectors and normal	Set true to generate Figure 1C	
N	Accepted xy distance of points for fit-width-smoothing (µm)	9.6	9.6
O	# (+/−)-consecutive points for xy- and fit-width-smoothing	15	15
P	Distance of point to first point to form the reference vector (µm)	10.0	6.4
Q	Curvature: reference point distance	10.0	10.0
R	FFT: Grouped consecutive time-steps	200	500
S	FFT: Do not analyze (frequency results of flagellar parameters for the) initial … µm from head	0.0	0.0
T	Head rotation matrix radius	10	10

**Table 2 cells-08-00010-t002:** Terminology and indications of SpermQ parameters.

Parameter	Indication
Arc length	Position on the flagellum.
Head position	Trajectory of the swimming path.
Head angle in space (Θ)	Main beat frequency, rolling frequency (freely-swimming sperm), orientation in space, curvature of the trajectory.
Maximum intensity in head (rolling)	Rolling frequency (freely-swimming sperm), main beat frequency.
x coordinate	Oscillations on the longitudinal axis through the head.
y coordinate	Beat amplitude, asymmetry of the whole beat waveform
Tangential angle	custom analysis approaches.
Curvature/curvature angle	Localization of the origin and propagation of waves on the flagellum, wave kinetics, beat frequency, local beat amplitude, local beat asymmetry.
z position	Qualitative value to describe the kinetics of the flagellar beat in the direction vertical to the focal plane.
Intensity	Intensity profile of the cell along the whole flagellum, instructive when the cell is labelled with a fluorescence indicator; intensity also depends on the position of the flagellum in z.

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
