# Peer review of "SpermQ–A Simple Analysis Software to Comprehensively Study Flagellar Beating and Sperm Steering"

_cells, 2018, doi:10.3390/cells8010010_

Round 1
Reviewer 1 Report
I have reviewed the manuscript by Hansen et al. and find that it will make a valuable contribution to the field. This new software module will extend the usefulness of sperm motility analysis beyond what is currently done for mammalian, and especially, human sperm analyses. The paper is written well and I have no major comments. Two minor comments are below:
1. As is typical of sperm analyses, the investigators have used dark field microscopy to collect images. As expected, this gives high contrast images but at the expense of resolution which could be important for 3D analyses. Have the investigators used other modes of microscopy such as DIC or phase contrast which would give better resolution of both the head and flagellum? If not, it might be worth adding something to the discussion to address this possibility. Of course, this might require additional image processing to increase contrast but I imagine that this would be relatively easy to achieve.
2. The use of a 500 frame per second camera is critical to accurate analysis of flagellar beating parameters. However, many investigators will not have access to these kinds of frame rates. Accurate determinations of flagellar frequency and amplitude require that the Nyquist condition be satisfied. A discussion of this point should be made in the discussion and, whenever possible, what aspects of SpermQ can be used with standard CASA rigs.
3. Since the 150 micron depth chamber is custom-made and, therefore, not typical of what is used by others, additional description (or a figure), with the actual design is warranted.
Author Response
We would like to thank the reviewers for their valuable and constructive criticism of the original manuscript. We addressed the issues raised and changed the manuscript accordingly.
Reviewer 1
1. As is typical of sperm analyses, the investigators have used dark field microscopy to collect images. As expected, this gives high contrast images but at the expense of resolution which could be important for 3D analyses. Have the investigators used other modes of microscopy such as DIC or phase contrast, which would give better resolution of both the head and flagellum? If not, it might be worth adding something to the discussion to address this possibility. Of course, this might require additional image processing to increase contrast but I imagine that this would be relatively easy to achieve.
The reviewer raised an important point. We would like to clarify the following points:
a) There is no major increase in resolution when applying DIC or phase contrast (PC) compared to dark-field microscopy. The resolution is mainly dependent on the objective applied. Of note, applying a high-magnification objective will reduce the field of view (FOV), which is disadvantageous when looking at freely swimming sperm. With our setup, applying a 16x magnification results in a pixel length of about 0.7 µm/pixel, whereas a 60x magnification would result in a pixel length of 0.18 µm/pixel. Mouse sperm are about 120 µm in length. Thus, we would need a FOV of at least 250 * 250 µm for tethered sperm that rotate around their tethered point. If we want to track freely swimming sperm for longer than one second, we would need a FOV of at least 1388 * 1388 pixel with a 60x magnification or 358 * 358 pixel with a 16x magnification. Since we need a high temporal resolution, we cannot record high SNR 200 fps image sequences on the full camera chip at a resolution of 1388 * 1388 pixel. However, reducing the recorded image size to the center of the camera chip and using only 358 * 358 pixel, we can achieve frame rates of 500 fps. Thus, there is a fine balance between high temporal and spatial resolution. In addition, we demonstrate that using SpermQ, even a “rather” low spatial resolution allows to reconstruct the flagellum with a high sub-pixel resolution by using Gaussian curve fits along the sperm cell. We do not see any reason why the resolution that our analysis provides (see Figures 3, 4) is not sufficient for investigating the flagellar beat of sperm in 3D.
b) We selected dark-field microscopy instead of DIC and PC microscopy because tracking of sperm is easier in dark-field microscopy. Of course, pre-processing of DIC or PC images may render the images suitable for SpermQ reconstruction and we do not want to exclude this possibility. Thus, we have included this point in the discussion and thank the reviewer for this helpful suggestion. However, we face the following problems when using DIC or PC images that render analysis more difficult:
1. In PC, halo artifacts occur – especially, when imaging “big” objects like the sperm head. This renders the reconstruction of the sperm head and the initial flagellum more complex and will affect the reconstruction method applied by SpermQ. Additionally, these halos will eventually cause gaps in the reconstructed flagellum, when performing image segmentation and skeletonization of the sperm flagellum. To perform flagellar reconstruction in SpermQ, PC data will require pre-processing.
2. We take advantage of the fact that in dark-field microscopy, the point-spread function can be approximated by a Gaussian curve. Thereby, we are able to achieve a sub-pixel resolution for the sperm flagellum. However, in PC or DIC, the point-spread function is much more complex (higher-order polynomial functions) and thus, would require a more complex and computationally more demanding fitting.
2. The use of a 500 frame per second camera is critical to accurate analysis of flagellar beating parameters. However, many investigators will not have access to these kinds of frame rates. Accurate determinations of flagellar frequency and amplitude require that the Nyquist condition be satisfied. A discussion of this point should be made in the discussion and, whenever possible, what aspects of SpermQ can be used with standard CASA rigs.
The reviewer is right. We have included in the discussion that an accurate determination of flagellar frequency is needed to satisfy the Nyquist-Shannon sampling theorem.
All our data were acquired with a PCO Dimax camera. This is of course a very expensive camera and might therefore not be accessible to all users. However, a cheaper camera, e.g. a Hamamatsu camera, also provided images that were well suited for SpermQ analysis, even with high frame rates of up to 500 Hz. Although resulting images show a lower SNR than those acquired with the PCO Dimax camera, Gaussian curve fits in SpermQ allow a high spatial resolution in the final analysis.
Standard CASA rigs analyze a dense sperm population using bright field-imaging (in some cases including phase contrast) at an acquisition rate of 50-200 fps. As discussed above, the input of PC or DIC images is possible, but requires more pre-processing and a more elaborate fitting regime. Furthermore, higher frame rates would be needed. We have included this in the discussion.
3. Since the 150 micron depth chamber is custom-made and, therefore, not typical of what is used by others, additional description (or a figure), with the actual design is warranted.
We have included a schematic drawing of the chamber (Fig. 1).
Reviewer 2 Report
The authors present a Java-based ImageJ plugin (SpermQ) to quantify 2D sperm motility from common time-lapse images acquired by dark-field or epifluorescence microscopy. The software determines the beat pattern, the intensity levels and the beating frequency along the flagellum, as well as the location and orientation of the cell. The software is able to analyze head-tethered as well as free-swimming sperm and the authors provide some examples of the typical analysis using mouse and human sperm time-lapse images. Finally, to visualize the results, an open-source java application is presented (SpermQ Evaluator) which is publicly available.
I find the software a very useful tool to characterize the spatiotemporal dynamics of flagellar beating. The software provides an automatized in-depth analysis of sperm time-lapse images which I believe can help clinicians to better quantify sperm motility. In addition, the software is also of high interest in the academic world, given that it is not trivial to obtain high-quality tracking of flagellar and ciliary beating.
Below, I enumerate a list of questions and comments that I think would significantly improve the readability and quality of the manuscript. In particular, I think there is a major issue on the clarity and readability of the mathematical expressions. Provided all my comments and questions are successfully addressed, I would recommend the publication of the manuscript in the journal Cells.
Major issue
I have a major concern regarding the notation used and the mathematical definitions throughout the manuscript. This specially applies to Section 2.5. The notation used is confusing and not concise and the format is not appropriate for common journal standards. Please use subscripts for indices and bold letters when referring to vectors. Below I specify some of the problems and questions I have regarding the mathematical definitions:
1) In the line 111, the authors explain the method they use to sort the list of points P corresponding to each pixel along the flagellum. Is |P| the size of the array? In case this is correct, do the authors look for the right order of the array P that minimizes the quantity sum_i=2^|P||p_i-p_(i-1)|? Is |p_i-p_(i-1)| the distance between the points p_i and p_(i-1)? If this is the case, please define the modulus. I would also write p_i in bold given that p_i is a vector defining the position of a pixel.
2) In the lines 122-127 the authors define the tangential and normal vectors t and n. Should not each point p_i have tangential and normal vectors associated? Then the right notation would be t_i and n_i. I would suggest using bold notation for vectors rather than an arrow.
3) In the line 125, the authors define the normal vector as n=t .(-1,1). The expression is incorrect since a dot product between two vectors gives a scalar and not a vector. I presume the authors impose the conditions (n_i . t_i)=0, n_i . n_i=1 and t_i . t_i=1. Then two solutions are possible (nxi,nyi)=(tyi,-txi) or (nxi,nyi)=(-tyi,txi), where t_i=(txi,tyi) and n_i=(nxi,nyi). Depending on the convention, the cross product of n_i and t_i, gives a vector pointing out of the plane or in the plane. This has implications on the sign of the curvature. The sentence included to define the choice of the sign is rather cumbersome: “The sign convention used for normal direction was such that if the cross product was positive, n was multiplied by -1”. Please see for example Ref. [Camalet & Jülicher, 2000] for a clear definition.
4) In line 128, the authors say “the normal vector n was used to generate a normal line […] from p_i towards both orthogonal directions compared to t”. I find “both orthogonal directions compared to t” confusing. I would rather say “at each point along the flagellum a line normal to the tangential direction was generated”.
5) In Fig. 2B and Fig. 4C, I do not understand what the subscript “i” stands for in Θ.
Questions and comments
1) I do not understand what “normal radius for gauss fit” means.
2) From line 144 to line 150, the authors explain their procedure to smoothen the flagellar track. I do not have a clear intuition why this method can smoothen the curve. In case this is a standard method, I would suggest the authors to include some references referring to the method. Otherwise, it would be nice if the authors could give some additional insights.
3) The authors classify the outputs of the software in two groups: ‘Orientation parameters’ and ‘Flagellar parameters’. I find this classification not precise since the the head position is not an orientation. I would rather change ‘orientation parameters’ for ‘head parameters’.
4) I believe that the tangential angle at each point along the arclength of the flagellum is a very important output to have (as well as the curvature). From the paper, I am not completely sure if this is a direct output of the plugin. If not, I think it would be important to add it since it is an essential quantity together with the head position and head orientation.
5) I do not understand the definition of the parameter in Table 2S.
6) I do not understand what “head rotation matrix radius” and “head rolling” mean. I think it would be helpful if these quantities are better explained in the text or in Fig. 2C.
7) In Fig. 2E, I presume the parameter “d” is chosen to be 5 µm, but is not clearly stated in the caption. Please clarify it.
8) In the caption of Fig. 2, the authors mention that “the initial 6 µm of the flagellum define the x-vector” or “The curvature angle at a given point is calculated by the angle between the tangential vector at the given point and the tangential vector of the point 10 µm before the given point”. As far as I understand these parameters are variable in the software and correspond to the ones in Table 2P and Table 2Q, respectively. From the text in the caption they seem to be fixed rather than variable. Please clarify it.
9) Why does Fig. 2 appear before Fig. 1 in the manuscript?
10) Figs. 1B and 1D look very similar to me. Is Fig. 1D the smoothened version of Fig. 1B?
11) In the line 255, the authors state “This approach has been proposed already in the 19th century” but then they cite a reference from 1960. Is this correct?
12) Are the sequence of images in Fig. 3B the same as in Fig. 3A? Given that the scale bar is the same in both cases, I find the flagellar shapes quite different.
13) In Fig. 3G (top), the primary frequency from ~40 to 100 µm in arclength is around ~20 Hz but in the bottom panel, the frequency in red is shown as 11-12Hz. Conversely, the secondary frequency is of ~10 Hz in the same range. How do these results compare with the analysis in Fig. 3E?
14) In my opinion, some important literature on flagellar beating is missing. I would include first some seminal works [Satir et al, 1968; Summers et al., 1971, Brokaw 1989], a general review [Gaffney et al., 2011], mention other studies that quantified flagellar beating in detail [Riedel-Kruse et al, 2007; Smith et al., 2009; Ooi et al., 2011, Ma et al., 2014] and maybe some references on the proposed mechanisms underlying axonemal beating [Brokaw 1972; Hines & Blum, 1978-1979; Lindemann, 1994; Camalet et al. 2000, Bayly et al. 2015; Sartori et al. 2016; Oriola et al. 2017].
Author Response
We would like to thank the reviewers for their valuable and constructive criticism of the original manuscript. We addressed the issues raised and changed the manuscript accordingly.
Reviewer 2
Major issues
1. In the line 111, the authors explain the method they use to sort the list of points P corresponding to each pixel along the flagellum. Is |P| the size of the array?
Yes. We rephrased the text accordingly: “… (the index ? indicates the position of the point in ?; 0<?≤|?|, while |P| defines the number of points in the list).”
In case this is correct, do the authors look for the right order of the array P that minimizes the quantity sum_i=2^|P||p_i-p_(i-1)|?
Yes, we stated this on page 4 (top paragraph): “Next, ? is sorted so that sum_i=2^|P||p_i-p_(i-1)| is minimal.”
Is |p_i-p_(i-1)| the distance between the points p_i and p_(i-1)?
Yes.
If this is the case, please define the modulus. I would also write p_i in bold given that p_i is a vector defining the position of a pixel.
We thank the reviewer for this comment and adapted the expression accordingly.
I would also write p_i in bold given that p_i is a vector defining the position of a pixel.
We adapted the text accordingly.
2. In the lines 122-127 the authors define the tangential and normal vectors t and n. Should not each point p_i have tangential and normal vectors associated? Then the right notation would be t_i and n_i. I would suggest using bold notation for vectors rather than an arrow.
We adapted the text accordingly.
3. In the line 125, the authors define the normal vector as n=t .(-1,1). The expression is incorrect since a dot product between two vectors gives a scalar and not a vector. I presume the authors impose the conditions (n_i . t_i)=0, n_i . n_i=1 and t_i . t_i=1. Then two solutions are possible (nxi,nyi)=(tyi,-txi) or (nxi,nyi)=(-tyi,txi), where t_i=(txi,tyi) and n_i=(nxi,nyi). Depending on the convention, the cross product of n_i and t_i, gives a vector pointing out of the plane or in the plane. This has implications on the sign of the curvature. The sentence included to define the choice of the sign is rather cumbersome: “The sign convention used for normal direction was such that if the cross product was positive, n was multiplied by -1”. Please see for example Ref. [Camalet & Jülicher, 2000] for a clear definition.
Indeed the described definition of the normal vector in the methods part was wrong. We have changed the text accordingly: “Next, for each point (p_i ) ⃑, the tangential vector (t_i ) ⃑ pointing towards (p_(i+1) ) ⃑ is determined. The tangential vector (t_i ) ⃑ is determined by the vector between the points located a given number of points (Table 2I, divided by two) before and after (p_i ) ⃑. At the start or the end of P, closer points are selected to form (t_i ) ⃑. The normal vector (n_i ) ⃑ is determined as (n_i ) ⃑=(-t_(i,y),t_(i,x) ) for (t_i ) ⃑=( t_(i,x),t_(i,y) ). If the cross-product (t_i ) ⃑×(n_i ) ⃑ was negative, (n_i ) ⃑ was multiplied by -1. At each point of the flagellum, a line normal to the tangential direction and of a defined length was generated (normal vector (n_i ) ⃑, Table 2J)”.
4. In line 128, the authors say “the normal vector n was used to generate a normal line […] from p_i towards both orthogonal directions compared to t”. I find “both orthogonal directions compared to t” confusing. I would rather say “at each point along the flagellum a line normal to the tangential direction was generated”.
We adapted the text accordingly.
5. In Fig. 2B and Fig. 4C, I do not understand what the subscript “i” stands for in Θ.
We apologize for not explaining i in the legend. i was dedicated to describe the frame. To avoid any confusion, we removed the subscript i from all figures containing Θ.
Questions and comments
1. I do not understand what “normal radius for gauss fit” means.
“Normal radius for gauss fit” describes the parameter in the software that defines the half-length of all normal lines that are constructed at each flagellar position. To further explain this parameter, we added more information to the parameter definition in the table: “Normal radius for gauss fit (µm) – defines the distance of the each flagellar point to the start and the end of the corresponding normal line = half-length of each normal line”.
2. From line 144 to line 150, the authors explain their procedure to smoothen the flagellar track. I do not have a clear intuition why this method can smoothen the curve. In case this is a standard method, I would suggest the authors to include some references referring to the method. Otherwise, it would be nice if the authors could give some additional insights.
We have explained this in more detail now.
3. The authors classify the outputs of the software in two groups: ‘Orientation parameters’ and ‘Flagellar parameters’. I find this classification not precise since the the head position is not an orientation. I would rather change ‘orientation parameters’ for ‘head parameters’.
We adapted the text accordingly.
4. I believe that the tangential angle at each point along the arc-length of the flagellum is a very important output to have (as well as the curvature). From the paper, I am not completely sure if this is a direct output of the plugin. If not, I think it would be important to add it since it is an essential quantity together with the head position and head orientation.
So far, the software output includes the curvature, but not the tangential angle. We thank the reviewer for this suggestion and included the tangential angle in the output of SpermQ – see also Figure 2.
5. I do not understand the definition of the parameter in Table 2S.
The parameter was included to exclude the head (= the initial points up to an arc length corresponding to the threshold set in the parameter of Table 2S) from FFT analysis for the results of flagellar parameters. Note that the list of flagellar points starts in the center of the head in SpermQ with the first points of the list still residing in the head. To further clarify this, we provide a more detailed description in the methods part, starting from “Optionally, the sperm head can be excluded from the frequency analysis of flagellar parameters. …. The parameter from Table 2S is set to 0.0 by default, whereby nothing is excluded, but it can optionally be increased by the user if the sperm head is not relevant for the user. Thereby, the analysis speed can be increased. We included this option into SpermQ because we also investigated sperm, where the head was completely immobilized. Without any movement, the FFT analysis does not give reliable results. Thus, we excluded the initial microns of the flagellar track from FFT analysis to speed up sperm analysis and exclude the head and initial flagellum from any further analysis. We adapted the text accordingly.
6. I do not understand what “head rotation matrix radius” and “head rolling” mean. I think it would be helpful if these quantities are better explained in the text or in Fig. 2C.
We have explained these two parameters in more detail in the text and figure legend. Furthermore, we renamed “head rolling” into “rolling”, because this labeling was indeed misleading: “Rolling” refers to the rotation of the entire sperm cell around its longitudinal axis but not of the head only.
7. In Fig. 2E, I presume the parameter “d” is chosen to be 5 µm, but is not clearly stated in the caption. Please clarify it.
We have adapted the figure legend as follows: “d can be set by the user (see Table 2Q) and, here, was set to 10 µm (taking the points 5 µm up- and downstream into account for calculating the curvature, Table 2Q).
8. In the caption of Fig. 2, the authors mention that “the initial 6 µm of the flagellum define the x-vector” or “The curvature angle at a given point is calculated by the angle between the tangential vector at the given point and the tangential vector of the point 10 µm before the given point”. As far as I understand these parameters are variable in the software and correspond to the ones in Table 2P and Table 2Q, respectively. From the text in the caption they seem to be fixed rather than variable. Please clarify it.
We agree with the reviewer and adapted the figure legend accordingly.
9. Why does Fig. 2 appear before Fig. 1 in the manuscript?
We focused on presenting the figures in the order of how they appear in the results part. However, we agree with the reviewer that this might be confusing. Thus, we took out the references to figures in the methods part and replaced them by references to the results text.
10. Figs. 1B and 1D look very similar to me. Is Fig. 1D the smoothened version of Fig. 1B?
Yes and we agree with the reviewer that they look very similar. This is due to the fact that the fine version of the flagellar points has sub-pixel resolution. When applying the same resolution for both images, they look the same. To improve the figure according to the reviewer’s suggestion, we sketched the flagellar track by a vector graphic overlaying the image and added a more detailed description to the figure legend: “(the precision of the track after correction is below the image resolution and thus, cannot be correctly displayed in the image)”.
11. In the line 255, the authors state “This approach has been proposed already in the 19th century” but then they cite a reference from 1960. Is this correct?
This is a mistake and has been corrected in the revised version of the manuscript (20th century).
12. Are the sequence of images in Fig. 3B the same as in Fig. 3A? Given that the scale bar is the same in both cases, I find the flagellar shapes quite different.
Yes, the two sequences are the same. As stated in the figure legend, the trace presented in Fig. 3B is the trace after aligning it to the head-midpiece axis, while Fig. 3A shows the raw data. We verified the raw data and did not find any mistake in generating the raw data. However, we noticed that the time legend is wrong. Both figures do not show overlays of the frames between 0 and 0.8 s, but between 0 and 0.08 s. We apologize for this typing mistake and adapted the figure accordingly.
13. In Fig. 3G (top), the primary frequency from ~40 to 100 µm in arc length is around ~20 Hz but in the bottom panel, the frequency in red is shown as 11-12Hz. Conversely, the secondary frequency is of ~10 Hz in the same range. How do these results compare with the analysis in Fig. 3E?
We apologize for this mistake and adapted the figure accordingly in line with Fig. 3E.
14. In my opinion, some important literature on flagellar beating is missing. I would include first some seminal works [Satir et al, 1968; Summers et al., 1971, Brokaw 1989], a general review [Gaffney et al., 2011], mention other studies that quantified flagellar beating in detail [Riedel-Kruse et al, 2007; Smith et al., 2009; Ooi et al., 2011, Ma et al., 2014] and maybe some references on the proposed mechanisms underlying axonemal beating [Brokaw 1972; Hines & Blum, 1978-1979; Lindemann, 1994; Camalet et al. 2000, Bayly et al. 2015; Sartori et al. 2016; Oriola et al. 2017].
The reviewer is right and we apologize for not citing this important literature. We have now included the missing citations.